# LoRE: Robust and Adaptive Graph Embeddings via Local Self-Reconstruction Mechanisms

## Abstract

Graph embeddings aim to project nodes into numeric vector spaces that capture structural and semantic regularities, enabling their use as general-purpose representations for a broad range of downstream applications. However, existing embedding methods distort local geometry through negative sampling, fail to enforce semantic consistency, and require expensive retraining when graphs evolve. Therefore, we introduce LoRE, a versatile graph embedding framework based on attention-driven self-reconstruction mechanisms and a perspective-preserving training procedure. Built on a generalized formulation, LoRE can be applied to a wide range of graph types, from undirected graphs to relational knowledge graphs and even attributed node sets without inherent topologies. It enforces identical embeddings for structurally equivalent nodes, respects local context during training, and reduces the likelihood of violations of the open-world assumption. Unlike traditional methods, LoRE supports efficient on-the-fly adaptation: embeddings can be updated in real time as graphs change, without full retraining. Its reconstruction mechanism acts as a self-supervised training signal that improves embedding robustness, yielding improved performance compared to existing approaches. Extensive experiments demonstrate that LoRE consistently matches or outperforms baseline results while maintaining stability under dynamic conditions. Qualitative analyses further show that LoRE produces more separable and compact clusters in embedding spaces. Together, the results underscore its enhanced generalizability and practical value as a global, task-agnostic embedding method.

## 1 Introduction

Graphs provide a natural framework for representing data with inherent relational structure. Given a set of nodes, they capture relationships between them as edges and may include additional metadata, such as edge directions, weights, or labels. Such representations are central to a wide range of domains, including biomedicine, finance, and manufacturing (Zhou et al., 2020). Knowledge graphs (KGs) further extend this paradigm by supporting logical inference through axioms encoded in ontologies, enabling the integration of statistical and symbolic reasoning (Hogan et al., 2021). To make graph-structured data usable in a task-agnostic manner, graph embedding (GE) methods map nodes into continuous vector spaces (Wang et al., 2017), enabling downstream machine learning tasks like node classification, link prediction, and graph-based recommendation (Nickel et al., 2016). These general-purpose GE methods are usually trained in a self-supervised manner and are commonly organized into four main categories: factorization-based approaches (Nickel et al., 2011; Ou et al., 2016), random-walk-based techniques (Ristoski & Paulheim, 2016; Grover & Leskovec, 2016), translational distance models (Bordes et al., 2013; Lin et al., 2015), and graph neural networks (GNNs) (Kipf & Welling, 2017; Veličković et al., 2018). Despite their successes, several practical requirements remain unmet. We identify four key properties essential for robust and adaptive GEs in real-world applications, reflecting challenges such as noise, consistency, and incompleteness:

P1 **Neighborhood-Invariance:** Structurally indistinguishable nodes yield identical embeddings.

P2 **OWA Obedience:** Training procedures avoid violating the open world assumption (OWA).

P3 **Locality-Awareness:** Local relationships must be preserved during the embedding process.

P4 **Graph Dynamics:** Embeddings must adapt to graph updates in real time without retraining.

These properties constitute critical requirements for practical implementations, yet, as we show in Section 2, current GE approaches do not consistently meet them. To overcome these challenges, we propose the LORE approach (LOcally Reconstructed Embeddings). It reconstructs randomly initialized base embeddings via attention mechanisms, using only the embeddings of immediate neighbors and their relation types as input. The resulting reconstructions are used as the final embeddings, enforcing neighborhood-invariance **(P1)** by ensuring that structurally identical nodes yield identical representations. Unlike traditional edge-level negative sampling, LORE draws node-level negatives for its multi-edge reconstructions of positive target node embeddings, thereby reducing the risk of violating the OWA **(P2)**. When combined with margin-ranking loss (MRL) using fixed-perspective distances, embeddings of plausibly similar nodes may be pushed apart. To avoid such distortions, LORE employs a perspective-preserving angular loss that respects relative similarities of positive and negative samples, maintaining local relationships while enforcing necessary separation during training **(P3)**. Finally, since embeddings are built from local context, LORE naturally adapts to dynamics through updated forward passes **(P4)**. Benchmark evaluations confirm that LORE not only outperforms existing baselines but also handles incremental updates arising from graph incompleteness, underscoring its potential as a robust and versatile solution for real-world graphs.

## 2 RELATED WORK

Graph embeddings (GEs) $\phi : \mathcal{V} \to \mathbb{R}^d$ transform graph nodes $\mathcal{V}$ into numerical vectors. However, real-world graphs are often incomplete, dynamic, and sparse (Xu, 2021; Krause et al., 2022), implying requirements for practical implementations of GEs, from which we formalize four key properties P1–P4. First, distinct nodes can share identical neighborhoods, especially in sparse graphs. Neighborhood-invariance **(P1)**, as per Definition 7, ensures that such nodes are treated consistently, which is essential for fair and stable predictions (Zemel et al., 2013), particularly in high-risk domains. Next, negative sampling is a common strategy for defining the training loss in self-supervised GE approaches, where edges are corrupted to create negative training targets (Kamigaito & Hayashi, 2022), typically combined with MRL to penalize negative facts. However, under OWA, the absence of an edge does not imply its incorrectness. Using corrupted facts as training objectives can therefore violate OWA in graphs, such as KGs (Hogan et al., 2021), by penalizing true but unobserved facts. OWA obedience **(P2)** requires that the training procedure minimizes the likelihood of such violations. Moreover, while computationally efficient, negative sampling can distort the local geometry of the embedding space if negative target nodes that are actually similar to the positive ones are pushed apart. Locality-awareness **(P3)** preserves similarities within the embedding space during training: negative target embeddings should be treated differently depending on their proximity to positive ones to avoid unnecessary distortion of local geometry. Finally, adaptability of embeddings to graph dynamics in real-time without retraining **(P4)** is essential when graphs contain noise and evolve over time, as is often the case for real-world graphs (Yang et al., 2024). Thus, P1–P3 define conditions under which an embedding can be considered robust, whereas P4 describes dynamic capabilities. We now review four major categories of GE methods with respect to these properties.

**Factorization-based approaches** approximate adjacency matrices $A$ of graphs using low-rank decompositions (Cui et al., 2017). For instance, RESCAL (Nickel et al., 2011) factorizes $A \approx XRX^\top$ into node embeddings $X$ via interaction matrices $R$. Since $X$ is randomly initialized and optimization noise can cause embeddings of structurally equivalent nodes to diverge, such methods do not fulfill P1. By treating unseen facts as zeros during optimization, they also violate P2, and they do not explicitly convey P3 since the objective is global reconstruction of the full adjacency matrices rather than preservation of local patterns. Furthermore, updates require retraining, violating P4.

**Translational models** are particularly designed for KGs to represent relations as translations in vector spaces. Given a triple $(v, u, r)$, which implies a directed edge of type $r$ from the source node $v$ to the target node $u$, translational models approximate $\phi(u)$ via $\phi(v)$ and $r$. Several approaches exist (Wang et al., 2017), such as TransE (Bordes et al., 2013), TransR (Lin et al., 2015), and TransD (Ji et al., 2015). For instance, TransE assumes $\phi(h) + \rho(r) \approx \phi(t)$ with auxiliary relation embeddings $\rho(r)$, effectively approximating $\phi(t)$. Random initialization and training noise cause these models to fail P1. Corrupted triples are treated as false, violating P2. MRL with fixed-perspective distances, such as $\|\cdot\|_2$, are used with negative sampling, which can distort local embedding structures, therefore P3 is not maintained. Finally, retraining is required for updates, so P4 is not addressed.

**Random-walk-based techniques** such as Node2Vec (Grover & Leskovec, 2016) and RDF2Vec (Ristoski & Paulheim, 2016) generate sequences of nodes (and labels for labeled graphs) via random walks and learn embeddings using word2vec-style objectives. Together with random initialization of embeddings, they therefore do not fulfill P1, and negative sampling of nodes or labels that have not co-occurred in a walk violates P2. These methods capture co-occurrence patterns via context windows, which indirectly preserve local neighborhoods and similarities (P3) (Portisch & Paulheim, 2024; Khoshraftar & An, 2024). In contrast, adapting to graph updates requires both recomputing random walks and retraining (Hahn & Paulheim, 2024), so P4 is not satisfied.

**Graph Neural Networks (GNNs)** such as Graph Convolutional Network (GCN) (Kipf & Welling, 2017) and Graph Attention Network (GAT) (Veličković et al., 2018) consider a base embedding $\phi : \mathcal{V} \to \mathbb{R}^d$ to produce aggregations $\phi' : \mathcal{V} \to \mathbb{R}^{d'}$ with respect to node neighborhoods $\mathcal{N}_{\mathcal{V}}(v) \subseteq \mathcal{V}$:

$$\phi'(v) = \sigma\big(W\,\phi(v) + \sum\nolimits_{u \in \mathcal{N}_{\mathcal{V}}(v)} \omega_{v,u} \cdot W\,\phi(u)\big). \tag{1}$$

Here, $W \in \mathbb{R}^{d' \times d}$ is a learnable matrix, $\omega_{v,u} \in \mathbb{R}$ are aggregation weights (uniform for GCNs and attention-based for GATs), and $\sigma$ is an activation function. Relational GNNs (RGNNs), such as RGCN (Schlichtkrull et al., 2017) and RGAT (Busbridge et al., 2019), extend GNNs through edge neighborhoods $\mathcal{N}_{\mathcal{E}}(v) \subseteq \mathcal{V} \times \mathcal{O} \times \mathcal{R}$ with orientations $\mathcal{O}$ and relations $\mathcal{R}$ as

$$\phi'(v) = \sigma\big(W\,\phi(v) + \sum\nolimits_{(u,o,r) \in \mathcal{N}_{\mathcal{E}}(v)} \omega_{v,(u,o,r)} \cdot W_{o,r}\,\phi(u)\big), \tag{2}$$

with orientation- and relation-specific matrices $W_{o,r}$. GNNs require self-supervised signals like edge reconstruction or contrastive objectives for training. They do not satisfy P1, since random initialization of the base GE $\phi$ can lead to divergent representations $\phi'$ for structurally identical nodes. P2 and P3 depend on the chosen training signal, while P4 is only partially addressed, as aggregations can be recomputed but retraining is required. Notably, Navi adapts GNNs by masking self-projections within the GNN forward pass (Krause, 2022). Moreover, for a node $v$, it is optimized to reconstruct its base embedding, meaning that base embeddings are approximated by neighborhood embeddings independently of themselves. While this ensures that P1 and P4 hold, Navi requires a fixed, pre-trained base embedding, which violates P2 and P3, making downstream performance depend on its quality. To overcome this limitation, we introduce LoRE, an end-to-end GE framework that extends the idea of local self-reconstructions and fulfills all properties P1–P4.

## 3 PRELIMINARIES & DEFINITIONS

In this section, we introduce the principles and definitions of graphs and their embeddings from a generalized perspective, introducing embeddings as continuous representations of attributed nodes.

### 3.1 GRAPHS

Graphs $\mathcal{G} = (\mathcal{V}, \mathcal{E}; \mathcal{M})$ provide a flexible formalism for representing nodes $\mathcal{V}$ and their relationships as edges $\mathcal{E} \subseteq \mathcal{V} \times \mathcal{V} \times \mathcal{M}$ enriched with edge metadata $\mathcal{M}$. Analogous to Navi, we adopt the convention that graphs contain no self-edges, i.e., $\mathcal{E} \subseteq \{(v, u, m) \in \mathcal{V} \times \mathcal{V} \times \mathcal{M} \mid v \neq u\}$. This assumption is made without loss of generality, since self-edges carry no contextual information and can be treated as non-contextual node attributes, as we show below. Further, each graph induces a simple graph by discarding edge metadata and retaining only its underlying connectivity.

**Definition 1.** *Given $\mathcal{M} = \emptyset$, a graph $\mathcal{G} = (\mathcal{V}, \mathcal{E}; \emptyset) =: (\mathcal{V}, \mathcal{E})$ is called a simple graph (SG) with edges $\mathcal{E} \subseteq \mathcal{V} \times \mathcal{V}$. Unless otherwise stated, SGs are directed, meaning $(v, u) \in \mathcal{E}$ is an edge from $v$ to $u$. Undirectedness is obtained by requiring $(v, u) \in \mathcal{E} \iff (u, v) \in \mathcal{E}$.*

SGs thus provide a minimal representation of graphs via pairwise links, directed or undirected. In practice, however, many graphs are enriched with edge labels to encode additional semantics.

**Definition 2.** *Given a set of labels $\mathcal{L}$, a graph $\mathcal{G} = (\mathcal{V}, \mathcal{E}; \mathcal{L})$ is called a labeled graph (LG), implying $\mathcal{E} \subseteq \mathcal{V} \times \mathcal{V} \times \mathcal{L}$. Directedness is defined as in SGs: $(v, u, \ell) \in \mathcal{E}$ denotes an edge with label $\ell \in \mathcal{L}$ from $v \in \mathcal{V}$ to $u \in \mathcal{V}$, and undirectedness is obtained via $(v, u, \ell) \in \mathcal{E} \iff (u, v, \ell) \in \mathcal{E}$.*

Each LG thus induces an SG by collapsing its edges along the labels to prevent duplicates. Among LGs, knowledge graphs form an important subclass where labels correspond to semantic relations.

**Definition 3.** *Given a set of relation type labels $\mathcal{R}$, a directed LG $\mathcal{G} = (\mathcal{V}, \mathcal{E}; \mathcal{R})$ is called knowledge graph (KG). Triples $(v, u, r) \in \mathcal{E}$ denote edges of type $r \in \mathcal{R}$ from $v \in \mathcal{V}$ to $u \in \mathcal{V}$.*

Besides contextual attributes, several real-world KGs additionally encode non-contextual node attributes such as strings or timestamps. Self-edges can be handled in the same way, as they carry no contextual information. While our evaluation does not incorporate non-contextual attributes, we introduce the generalized notion of attributed node sets to show LoRE's applicability.

**Definition 4.** *Given a node set $\mathcal{V}$ and a non-empty attribute domain $\mathcal{D}_\mathcal{A}$, an attributed node set is defined as $\mathcal{S} = (\mathcal{V}, \mathcal{A})$ with an attribute assignment $\mathcal{A} \subseteq \mathcal{V} \times \mathcal{D}_\mathcal{A}$ and the characteristic mapping*

$$c_\mathcal{A} : \mathcal{V} \to 2^{\mathcal{D}_\mathcal{A}} \setminus \{\emptyset\}, \qquad c_\mathcal{A}(v) = \{ a \mid (v, a) \in \mathcal{A} \},$$

*where $2^{\mathcal{D}_\mathcal{A}} = \{ X \subseteq \mathcal{D}_\mathcal{A} \}$ is the power set of $\mathcal{D}_\mathcal{A}$ and $c_\mathcal{A}(v)$ is the collection of attributes of $v \in \mathcal{V}$. Moreover, this implies that every node $v \in \mathcal{V}$ has at least one attribute.*

This definition formalizes how attributes are associated with nodes without imposing any specific topology. To enable adjacency-based self-reconstructions, LoRE requires contextual attributes per node $v \in \mathcal{V}$. Accordingly, we define graphs as special cases of attributed node sets.

**Definition 5.** *Let $\mathcal{S} = (\mathcal{V}, \mathcal{A})$ be an attributed node set and let $\mathcal{M}$ be a possibly empty set of available edge metadata. Then, $\mathcal{S}$ is equivalent to a graph $\mathcal{G} = (\mathcal{V}, \mathcal{E}; \mathcal{M})$ if and only if*

$$\mathcal{D}_\mathcal{A} = \mathcal{V} \times \mathcal{M}, \quad and \quad c_\mathcal{A}(v) \subseteq \{(u, m) \in \mathcal{V} \times \mathcal{M} \mid u \neq v\} \text{ for all } v \in \mathcal{V}$$

*holds, i.e., $\mathcal{E} = \mathcal{A}$. Under this convention, directions must be explicitly encoded within edge metadata. Otherwise, incoming and outgoing edges are indistinguishable. Moreover, we say that $\mathcal{S}$ induces $\mathcal{G}$ if and only if $\mathcal{A}' \subseteq \mathcal{A}$ exists such that $\mathcal{S}' = (\mathcal{V}, \mathcal{A}')$ is equivalent to $\mathcal{G}$.*

As specified by Definition 5, we treat graph edges as attributes, where each attribute corresponds to a link to another node and may carry metadata such as relation types or timestamps, thereby unifying common graph structures. Moreover, it allows for extending contextual graph structures with non-contextual node attributes. To be more precise, an undirected SG is obtained by selecting $\mathcal{M} = \emptyset$ so that $u \in c_\mathcal{A}(v)$ identifies an edge between $v$ and $u$. To represent directedness, we include orientational attributes $\mathcal{O} = \{o_{in}, o_{out}\}$ and set $\mathcal{M} = \mathcal{O}$ so that $(u, o_{out}) \in c_\mathcal{A}(v)$ holds for a directed edge from $v$ to $u$, implying $(v, o_{in}) \in c_\mathcal{A}(u)$. LGs are defined analogously using labels $\mathcal{L}$, i.e., $\mathcal{M} = \mathcal{L}$. In particular, KGs are also covered by choosing $\mathcal{M} = \mathcal{O} \times \mathcal{R}$ with relations $\mathcal{R}$ so that $(u, o_{out}, r) \in c_\mathcal{A}(v)$ and $(v, o_{in}, r) \in c_\mathcal{A}(u)$ holds for a directed edge of type $r \in \mathcal{R}$ from $v$ to $u$.

## 3.2 GRAPH EMBEDDINGS

Analogous to the introduction of graphs, we define their embeddings from a generalized perspective.

**Definition 6.** *Let $\mathcal{S} = (\mathcal{V}, \mathcal{A})$ be an attributed node set and let $\phi : \mathcal{V} \to \mathbb{R}^d$ be an embedding of dimension $d \in \mathbb{N}$. If $\mathcal{S}$ induces a graph $\mathcal{G} = (\mathcal{V}, \mathcal{E}; \mathcal{M})$, $\phi$ is also called graph embedding (GE) of $\mathcal{G}$, and $\phi(v)$ represents the embedding of $v \in \mathcal{V}$. In case a projection $\pi : 2^\mathcal{A} \to \mathbb{R}^d$ exists such that*

$$\phi(v) = \pi(c_\mathcal{A}(v))$$

*holds, then $\phi$ is called an attributed embedding with respect to the attributes assigned via $\mathcal{A}$.*

Some works use the term graph embedding to denote a single vector for an entire graph. However, in this paper we refer to the collection of node embeddings within a graph. Moreover, we focus on real-valued embeddings in $\mathbb{R}^d$, though the LoRE approach naturally extends to other normed vector spaces, such as complex-valued embeddings (Trouillon et al., 2016; Sun et al., 2019). In standard practice, node embeddings are initialized randomly per node and optimized afterwards, introducing noise that may not be fully corrected during training. To ensure consistency and specify the key property of neighborhood-invariance for contextual GEs (P1), we formalize the requirement that nodes with identical context (e.g., identical neighborhoods) map to identical embeddings.

**Definition 7.** *Let $\mathcal{S} = (\mathcal{V}, \mathcal{A})$ be an attributed node set with a node embedding $\phi : \mathcal{V} \to \mathbb{R}^d$. We call the embedding attribute-invariant if and only if identical attributes yield identical embeddings:*

$$c_\mathcal{A}(v) = c_\mathcal{A}(u) \quad \Rightarrow \quad \phi(v) = \phi(u).$$

*To capture the idea that structurally identical nodes cannot be distinguished by their embeddings, we refer to attribute-invariant GEs as neighborhood-invariant in the remainder of this work.*

Attributed GEs are neighborhood-invariant by definition and therefore satisfy property P1. Navi showed that single-layer GNNs can also be neighborhood-invariant in principle when self-projections are removed. In practice, however, standard GNNs violate this property because their message-passing updates include node-specific self-projections (Equation 1, Equation 2).

# 4 METHODOLOGY

In this section, we introduce LoRE, which extends translational GE paradigms with attentive, GNN-style local self-reconstructions. LoRE enables end-to-end training of base embeddings and integrates the self-reconstruction constraint directly into the learning objective, thereby mitigating the limitations of strict one-hop modeling. By relying on translational objectives, LoRE also reduces the number of trainable parameters, resulting in a lean, robust, and adaptive GE framework.

## 4.1 LOCAL ATTRIBUTE AGGREGATIONS VIA LoRE

Given an attributed node set $\mathcal{S} = (\mathcal{V}, \mathcal{A})$, LoRE yields an attributed embedding $\psi : \mathcal{V} \to \mathbb{R}^d$, i.e., $\psi(v) = \pi(c_{\mathcal{A}}(v))$. Thus, a projection $\pi : 2^{\mathcal{D}_{\mathcal{A}}} \to \mathbb{R}^d$ as in Definition 6 is required. Intuitively, $\pi$ aggregates the information encoded in a node's attributes into a single representation. To achieve this, LoRE implements $\pi$ using a self-attention pooling mechanism over intermediate attribute embeddings $\{\Phi(a) \mid a \in c_{\mathcal{A}}(v)\}$ produced by a mapping $\Phi : \mathcal{D}_{\mathcal{A}} \to \mathbb{R}^d$. The attention weights determine how strongly each attribute contributes to the final representation, enabling the model to emphasize informative attributes for a predictive objective. Formally, the projection $\pi$ is defined as

$$\pi(A) = \sigma\left( \frac{1}{|A|} \sum_{a \in A} \left( \sum_{a' \in A} \omega_{a,a'} \, \Phi(a') \right) \right) = \sigma\left( \sum_{a \in A} \left( \sum_{a' \in A} \frac{\omega_{a',a}}{|A|} \right) \Phi(a) \right), \quad (3)$$

where $A \in 2^{\mathcal{D}_{\mathcal{A}}}$ is a finite attribute set, $\sigma : \mathbb{R}^d \to \mathbb{R}^d$ is an activation function, and $\Phi(a) \in \mathbb{R}^d$ is the intermediate representation of attribute $a \in A$. The intermediate embeddings $\Phi(a)$ are used to generate the attention weights: each attribute is mapped to a query $q_a = W_q \Phi(a)$ and a key $k_a = W_k \Phi(a)$, where $W_q, W_k \in \mathbb{R}^{d \times d}$ are learnable matrices. Scaled dot-products compute pairwise similarities, and a row-wise softmax yields

$$\omega_{a,a'} = \left( \sum_{a'' \in A} \exp\left( \frac{q_a^\top k_{a''}}{\sqrt{d}} \right) \right)^{-1} \cdot \exp\left( \frac{q_a^\top k_{a'}}{\sqrt{d}} \right).$$

As shown in Equation 3, the inner sum $\sum_{a' \in A} \omega_{a,a'} \Phi(a')$ is the self-attention update for each attribute $a$, obtained as a weighted combination of all intermediate attribute embeddings. The outer average then pools these updated representations into a single vector summarizing the entire attribute set, after which the activation function is applied. Equivalently, the projection can be viewed as a weighted sum of the intermediate embeddings $\Phi(a)$, where each weight reflects the relevance assigned to $a$ relative to all other attributes in $A$, mirroring the aggregation principle in GNN layers.

## 4.2 LOCAL GRAPH EMBEDDING RECONSTRUCTIONS VIA LoRE

To enable LoRE's neighborhood-invariant self-reconstruction objective, we assume that $\mathcal{S}$ is equivalent to a graph $\mathcal{G} = (\mathcal{V}, \mathcal{E}; \mathcal{M})$, so that each attribute $a = (u, m) \in c_{\mathcal{A}}(v)$ corresponds to a neighbor $u \neq v$ with optional edge metadata $m \in \mathcal{M}$. However, LoRE extends analogously to additional non-contextual attributes, provided their intermediate embeddings are well defined. Depending on the graph type, different implementations of $\Phi$ are possible. We restrict $\mathcal{M}$ to orientations and labels, with other edge metadata left for future work. Moreover, we adopt TransE's embedding paradigm for KGs as directed and labeled graphs with $\mathcal{M} = \mathcal{O} \times \mathcal{R}$ due to its simplicity and minimal number of relation parameters compared to relational weight matrices in GNNs. For $a = (u, o, r)$, we define

$$\Phi(a) = \phi(u) + \xi(o) \cdot \rho(r) \quad \text{with} \quad \xi(o) = \mathbb{1}_{\{o = o_{in}\}} - \mathbb{1}_{\{o = o_{out}\}} \in \{-1, 1\},$$

where both $\phi : \mathcal{V} \to \mathbb{R}^d$ (base GE) and $\rho : \mathcal{R} \to \mathbb{R}^d$ (auxiliary relation embedding) are learnable. For undirected LGs with $a = (u, r) \in \mathcal{V} \times \mathcal{R}$, we use $\Phi(a) = \phi(u) + \rho(r)$. For SGs, a single translational vector $t \in \mathbb{R}^d$ is used: if the graph is directed, we set $\Phi(a) = \phi(u) + \xi(o) \cdot t$ for $a = (u, o) \in \mathcal{V} \times \mathcal{O}$, while for undirected SGs this reduces to $\Phi(a) = \phi(u) + t$ with $a = u \in \mathcal{V}$.

Accordingly, LoRE's neighborhood-invariant embedding is defined as GNN-style aggregations

$$\psi(v) = \pi(c_{\mathcal{A}}(v)) = \tanh\left(\sum_{a \in c_{\mathcal{A}}(v)} \left(\sum_{a' \in c_{\mathcal{A}}(v)} \frac{\omega_{a',a}}{|c_{\mathcal{A}}(v)|}\right) \Phi(a)\right). \qquad (4)$$

By optimizing $\phi$, $\rho$, $W_q$, and $W_k$ with the reconstructive objective $\psi(v) \approx \phi(v)$, $\psi(v)$ becomes an independent self-reconstruction of $\phi(v)$, as $\phi(v)$ is not used in the forward pass of $v$. In this setup, $\phi(v)$ acts both as the reconstruction target for node $v$ and as contextual input for adjacent nodes. This decoupling allows $\psi(v)$ to rely purely on contextual information, while $\phi(v)$ is shaped through the reconstruction objective, as indicated by the results in Section 5. The base GE $\phi$ is randomly initialized and $\ell^2$-normalized throughout training, i.e., $\|\phi(v)\|_2 = 1$. The hyperbolic tangent activation bounds $\psi(v)$ within $(-1, 1)^d$, keeping it consistent with the unit-normalized $\phi(v)$ while preserving reconstruction flexibility and maintaining smooth gradients. In the following, we describe the optimization procedure of LoRE including a perspective-preserving training objective designed to address OWA-consistent negative sampling (P2) and locality-awareness (P3).

## 4.3 TRAINING LoRE

Most GE methods rely on edge-level negative sampling. Given a node $v \in \mathcal{V}$ and one of its contextual attributes $a = (u, m) \in c_{\mathcal{A}}(v)$, a negative node $\tilde{v}$ is sampled under the assumption that $a \notin c_{\mathcal{A}}(\tilde{v})$ implies a false fact. Training then effectively becomes a single-edge reconstruction task with input $a$, aimed at minimizing the distance to $\phi(v)$ while increasing the distance to $\phi(\tilde{v})$. Under the OWA, however, this assumption is brittle, since the graph may simply be missing the corresponding edge, thus violating P2. In contrast, LoRE follows Navi and reconstructs from the entire neighborhood $c_{\mathcal{A}}(v)$ rather than a single attribute. This reduces the risk of OWA violations because two nodes are far less likely to share all attributes than to share a single one. Unlike Navi, which keeps base GEs $\phi$ fixed, LoRE learns them end-to-end. Thus, for a node $v$ with base embedding $\phi(v)$ and self-reconstruction $\psi(v)$, we sample any $\tilde{v} \neq v$ and use $\phi(\tilde{v})$ as a negative target under a distance-based loss. For instance, cosine-based losses are implied by the cosine distance $\delta_{\cos} : \mathbb{R}^d \times \mathbb{R}^d \to \mathbb{R}$, which captures the relative orientation between vectors $x, x' \in \mathbb{R}^d$:

$$\delta_{\cos}(x, x') = 1 - (x \cdot x') \cdot (\|x\|_2 \cdot \|x'\|_2)^{-1} \in [0, 2].$$

In contrast to $\ell^p$ distances, which are translation-invariant but sensitive to scaling, cosine distance is invariant under positive scalar multiplication but not translation-invariant. To incorporate both translation- and scale-invariance, thereby enabling locality-awareness, we define $\delta_{\text{LoRE}}$ as a perspective-preserving cosine distance for a prediction $\hat{y}$ and the positive/negative target $y, \tilde{y}$:

$$\delta_{\text{LoRE}}(\hat{y}, y; \tilde{y}) = \delta_{\cos}(\hat{y} - \bar{y}, y - \bar{y}) \quad \text{with} \quad \bar{y} := (y + \tilde{y})/2.$$

The point $\bar{y}$ serves as a local anchor that re-centers the prediction and the positive target with respect to the negative target before measuring their angular difference. This adjustment allows the distance to be scale-invariant and locally translation-invariant relative to the anchor, as illustrated in Figure 1a.

Standard MRL optimizations compare two distances and enforce a margin $\mu \geq 0$ between them. Even though LoRE considers a single distance relative to the negative target, a margin $\mu \geq 0$ can still be specified. This results in the setup depicted in Figure 1b, i.e., the margin $\mu$ corresponds to an angle $\gamma_\mu$ and predictions $\hat{y}$ with $\gamma_{\hat{y}} > \gamma_\mu$ are penalized. Given an angle $\gamma_\mu \in [0, \frac{\pi}{2})$, the corresponding margin $\mu$ can be determined via $\mu = 1 - \cos(\gamma_\mu)$. Due to the $\ell^2$ normalization, the base embeddings $\phi$ are positioned on the unit sphere so the anchor $\bar{y}$ lies inside the unit ball. Thus, to account for locality-awareness as key property P3, LoRE is trained by minimizing the loss

$$\lambda_{\text{LoRE}}(v, \tilde{v}) = \max\left(\delta_{\text{LoRE}}(\psi(v), \phi(v); \phi(\tilde{v})) - \mu, 0\right). \qquad (5)$$

Experiments show that $\lambda_{\text{LoRE}}$ even improves downstream performance in the benchmark tasks of Section 5 compared to established alternatives by increasing predictive stability (Appendix A.3.3). Two strategies are applied to improve training efficiency and generalization. First, batching is applied, i.e., a threshold $\tau_b \in \mathbb{N}$ specifies the number of nodes processed together with their corresponding negative samples. Second, $\tau_a \in \mathbb{N}$ limits the number of attributes per node. If $|c_{\mathcal{A}}(v)| > \tau_a$, we construct a reduced attribute set $c'_{\mathcal{A}}(v) \subset c_{\mathcal{A}}(v)$ with $|c'_{\mathcal{A}}(v)| \leq \tau_a$ by randomly selecting at most $\tau_a$ attributes. To prevent overfitting to low-degree nodes, sampling is also applied when $|c_{\mathcal{A}}(v)| \leq \tau_a$, and the remaining $\tau_a - |c'_{\mathcal{A}}(v)|$ entries are masked in the attention mechanism.

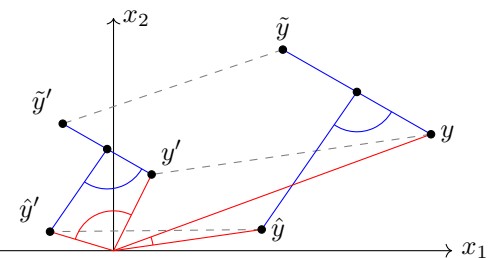 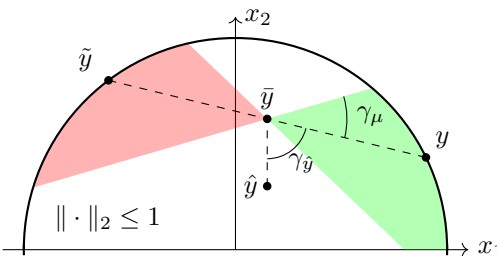

(a) Visual representation of the translation- and scale-invariance of $\delta_{\text{LoRE}}$ (resulting in its locality-awareness) as a perspective-preserving distance (blue), compared to regular cosine distance (red).

(b) Geometric interpretation of the $\delta_{\text{LoRE}}$-induced loss, relative to the target anchor $\bar{y}$. A prediction $\hat{y}$ is penalized if its angle $\gamma_{\hat{y}}$ to $y$ exceeds $\gamma_{\mu}$, thus preserving the perspective from the negative target $\tilde{y}$.

Figure 1: Visual representation and interpretation of $\delta_{\text{LoRE}}$ characteristics and training objectives.

## 5 EXPERIMENTAL EVALUATION

LoRE's embedding quality is evaluated on established GE benchmark tasks. We focus on node classification tasks, i.e., predicting externally defined labels from pretrained embeddings. We therefore restrict our evaluation to external data mining tasks rather than internal link prediction. While both aim to predict missing information (Portisch et al., 2022), external labels provide a stable, verifiable ground truth, whereas link prediction depends on edge-level negative samples that can violate the OWA. Simple undirected graphs and KGs are considered as representative graph types. In each setting, we compare LoRE against established baselines, demonstrating that it satisfies the properties P1–P4 and matches or outperforms existing methods, while its neighborhood-invariance prevents deceptive overperformances. Analogous to standard GE benchmarks, embeddings are first trained self-supervised (labels are withheld), after which support vector machines (SVMs) and random forests (RFs) predict node labels. Embedding creation thus serves as a pretraining step. Additional analysis indicates that LoRE's self-reconstruction mechanism improves embedding generalization, explaining the observed gains. Finally, we evaluate its dynamic capabilities (P4), showing it surpasses Navi and GNNs in settings that reflect dynamics caused by incompleteness.

LoRE embeddings of dimension $d = 256$ are trained using Adam with a fixed learning rate of $5 \cdot 10^{-5}$, a batch size of $\tau_b = 32$, and a maximum attribute count $\tau_a = 16$. The $\delta_{\text{LoRE}}$ angle margin $\gamma_{\mu}$ is selected from $\left\{0, \frac{\pi}{6}\right\}$ via grid search (i.e., 0° or 30°), and a dropout rate of 0.2 is applied. Downstream SVM hyperparameters are also selected via grid search over regularization values $C \in \{0.01, 0.1, 1, 10, 100\}$. For RFs, the number of trees $\{100, 300, 500\}$ and maximum depths $\{10, 30, 50\}$ are considered. GEs and classifiers are trained independently ten times, and results are averaged. Experiments were run on an NVIDIA RTX 2080 Ti GPU with 12 GB VRAM. To ensure reproducibility, the PyTorch implementation of LoRE is publicly available[1].

### 5.1 EMBEDDING UNDIRECTED SIMPLE GRAPHS

We evaluate on the standard Planetoid citation networks Cora, CiteSeer, and PubMed (Yang et al., 2016) and the Amazon-Computers co-purchase graph from the SNAP benchmark (Yang & Leskovec, 2012)as undirected SGs, where nodes are papers or products and edges are citations or co-purchases. The corresponding node classififcation benchmark tasks are widely used for evaluating the quality of GEs. Dataset characteristics are summarized in Table 4 (Appendix A.1). We adopt the fixed Planetoid and SNAP splits for comparability with prior work. Baselines include Node2Vec, GCN, and GAT (see Section 2), as well as HOPE (Ou et al., 2016) with training configurations given in Appendix A.2. For completeness, we also include GRACE (Zhu et al., 2020) as a strong contrastive benchmark. Mean accuracies and standard deviations are reported in Table 1, and macro-F1 scores in Table 6 (Appendix A.3.1) confirm the accuracy trends. GAT, GRACE, and LoRE achieve the highest accuracies, with LoRE exhibiting the lowest deviations and thus the most stable overall performance. GRACE required the longest training time, whereas HOPE and

---

[1]Anonymous repository with flask implementation: https://github.com/LoRE-ICLR/LoRE

Node2Vec trained fastest but performed worst. Per layer, LoRE is slightly slower and uses more parameters than a single GCN or GAT layer due to its attention mechanism; however, its single-layer design outperforms single-layer GNNs, and the GNN models required at least two layers to reach their best performance. Compared to multi-layer GNNs, LoRE is faster and more parameter-efficient overall. An efficiency analysis is provided in Section 5.3.

|          | HOPE         | Node2Vec     | GCN          | GAT          | GRACE        | LoRE         |
|----------|--------------|--------------|--------------|--------------|--------------|--------------|
| **Cora**     | 81.73 ±1.62 | 82.21 ±1.48 | 83.18 ±1.66 | **85.11** ±2.05 | 84.72 ±1.34 | 84.88 ± **0.95** |
| **CiteSeer** | 70.88 ±1.90 | 72.15 ±1.74 | 71.79 ±1.81 | 73.40 ±2.12 | **73.41** ±**0.96** | 73.12 ±1.04 |
| **PubMed**   | 79.64 ±1.72 | 79.12 ±1.55 | 80.73 ±1.43 | 81.95 ±1.63 | **83.01** ±1.17 | 82.21 ±**1.14** |
| **Amazon**   | 85.91 ±1.81 | 86.74 ±1.44 | 87.62 ±1.71 | 88.18 ±1.13 | 88.44 ±0.99 | **88.90** ±**0.96** |

Table 1: Mean accuracies including standard deviations for the SG embedding benchmark tasks.

## 5.2 EMBEDDING KNOWLEDGE GRAPHS

We use the benchmark KGs AIFB, MUTAG, BGS, and AM, which are widely adopted for evaluating KG embedding quality (Ristoski et al., 2016; Schlichtkrull et al., 2017). Each dataset provides a KG with literals and a predefined train–test split of node URIs and external class labels. Within our experiments, boolean literals are converted into instance–class relationships introducing two new classes per boolean property (true and false), while non-boolean literals are removed to match the contextual KG definition in Section 3. Dataset characteristics are summarized in Table 5 (Appendix A.1). Baselines are introduced in Section 2 and include RESCAL, TransE, TransR, TransD, RDF2Vec, RGCN, and RGAT, with detailed training configurations being described in Appendix A.2. Mean accuracies with standard deviations are shown in Table 2. Again, macro-F1 scores in Table 7 (Appendix A.3.1) confirm the observed accuracy trends.

|          | RESCAL       | TransE       | TransR       | TransD       | RDF2Vec      | RGCN         | RGAT         | LoRE         |
|----------|--------------|--------------|--------------|--------------|--------------|--------------|--------------|--------------|
| **AIFB**  | 84.10 ±1.28 | 87.22 ±1.63 | 85.56 ±1.72 | 71.39 ±1.88 | 88.89 ±1.77 | 91.30 ±1.22 | **92.22** ±1.53 | 91.67 ±**0.00** |
| **MUTAG** | 70.50 ±1.52 | 75.00 ±1.81 | 69.44 ±1.74 | 67.50 ±1.96 | 73.68 ±1.83 | 75.29 ±1.48 | 83.97 ±1.08 | **85.29** ±**0.91** |
| **BGS**   | 67.80 ±1.67 | 69.31 ±1.89 | 68.28 ±1.92 | 69.31 ±1.85 | 68.28 ±1.94 | 72.41 ±1.39 | 73.45 ±1.11 | **74.14** ±**1.03** |
| **AM**    | 71.25 ±1.63 | 74.75 ±1.79 | 73.79 ±1.84 | 79.95 ±1.91 | 87.47 ±1.68 | 91.26 ±1.26 | 92.42 ±**0.88** | **92.93** ±0.96 |

Table 2: Mean accuracies including standard deviations for the KG embedding benchmark tasks.

Once more, RGAT and LoRE achieve the highest accuracies overall, with LoRE slightly outperforming RGAT on MUTAG, BGS, and AM. However, the apparent RGAT lead on AIFB is not meaningful since ten of the 36 test nodes are topologically indistinguishable, and by design LoRE's neighborhood-invariance produces identical embeddings for such nodes, achieving the theoretical maximum accuracy of 91.67% by predicting the majority class. RGAT's higher score results from random variation and does not reflect a genuine advantage, as further training brings RGAT to the same value. Beyond these quantitative results, qualitative inspection of the embeddings confirms LoRE's improved separability over the baselines throughout the experiments. Appendix A.4, Figure 3, exemplifies this for the AIFB task, where LoRE produces distinctly separated and compact clusters compared to TransE and RGAT. Moreover, unlike RGCN and RGAT, which store a full projection matrix for each relation, LoRE represents each relation as a vector. Consequently, as the number of relations increases, LoRE becomes increasingly efficient in terms of training parameters and memory. This effect is examined in detail in the next section from a generalized perspective.

## 5.3 EMBEDDING EFFICIENCY AND DYNAMICS

We analyze and simulate the memory efficiency and scalability of LoRE's self-reconstruction layers by comparing them to GCN and GAT as representative GNN layers. We consider a randomly generated graph $\mathcal{G}$ with nodes $\mathcal{V}$ and relation types $\mathcal{R}$, setting $|\mathcal{R}| = 1$ for unlabeled graphs. All methods require $|\mathcal{V}| \cdot d$ base embedding parameters. For GNNs, each additional layer introduces at least $|\mathcal{R}| \cdot d^2$ parameters for relation-specific projection matrices. In contrast, LoRE represents relations as vectors and, together with its query and key matrices for self-attention, we obtain

$$\text{param}(\text{GNN}) \geq |\mathcal{V}| \cdot d + |\mathcal{R}| \cdot d^2 \qquad \text{and} \qquad \text{param}(\text{LoRE}) = (|\mathcal{V}| + |\mathcal{R}|) \cdot d + 2d^2.$$

Solving $\text{param}(\text{GNN}) \geq \text{param}(\text{LoRE})$ yields $|\mathcal{R}| \geq 2 \cdot \frac{d}{d-1}$, meaning that for typical dimensions such as $d = 256$, two relation types already yield similar parameter counts. Every additional

relation type further increases the parameter demand of conventional GNNs relative to LORE. Accordingly, forward and backpropagation times for random directed graphs scale steeply for GCN and GAT but remain nearly flat for LORE (see also Figure 2, Appendix A.3.2). Due to its single-layer architecture, LORE's efficiency advantage widens even further in multi-relational settings.

To evaluate dynamic adaptability (property P4), we again use the KG benchmarks and compare LORE with RGCN, RGAT, and Navi. Incremental updates are simulated by randomly removing roughly half of each test node's edges before training, while ensuring at least one edge remains. This controlled perturbation reflects common real-world scenarios where KGs are incomplete and new links are incrementally discovered. After embedding generation and classifier training, the removed edges are reinserted, and the models are evaluated again by computing updated embeddings and applying the same classifier. Each experiment is repeated ten times with identical deletions across models to ensure strict comparability. As shown in Table 3, RGAT and LORE initially perform better than RGCN and Navi. After reinsertion, RGCN and RGAT typically degrade, whereas Navi and LORE improve, with LORE consistently outperforming Navi across datasets. These results indicate that LORE's reconstruction mechanism supports stable, real-time embedding adaptation without costly retraining. A more extensive evaluation involving new nodes, node deletions, or modified relation types is out of scope for this evaluation and is left for future work.

| | RGCN Start | RGAT Start | Navi Start | LORE Start | RGCN End | RGAT End | Navi End | LORE End |
|---|---|---|---|---|---|---|---|---|
| **AIFB** | $59.10 \pm 4.87$ | $74.27 \pm 4.08$ | $72.57 \pm$ **3.54** | **76.03** $\pm 3.96$ | $55.44 \pm 3.63$ | $73.86 \pm 4.72$ | $75.17 \pm$ **1.17** | **85.20** $\pm 1.33$ |
| **MUTAG** | $61.59 \pm 5.26$ | **65.17** $\pm 4.67$ | $60.44 \pm$ **3.22** | $63.60 \pm 3.48$ | $57.94 \pm 4.98$ | $66.02 \pm 4.92$ | $65.04 \pm$ **0.89** | **66.18** $\pm 1.14$ |
| **BGS** | $52.30 \pm 5.81$ | **69.40** $\pm 5.03$ | $69.03 \pm 3.85$ | $68.20 \pm$ **2.26** | $49.75 \pm 5.39$ | $63.14 \pm 5.58$ | $71.58 \pm 1.21$ | **72.36** $\pm$ **0.97** |
| **AM** | $61.01 \pm 4.33$ | **62.18** $\pm 4.41$ | $55.23 \pm$ **3.37** | $60.79 \pm 3.59$ | $60.80 \pm 4.44$ | $59.96 \pm 4.18$ | $58.50 \pm$ **1.24** | **65.19** $\pm 1.19$ |

Table 3: Mean accuracies including standard deviations before (Start) and after (End) incompleteness simulations and real-time embedding adaptations with updated neighborhood information.

## 6 CONCLUSION

In this work, we introduced LORE, a global and task-agnostic graph embedding framework that can be adapted to a wide range of graph types, making it a versatile choice for diverse domains. It is designed to satisfy the four key properties P1 to P4 defined in this paper. Its local self-reconstruction mechanism enforces neighborhood invariance (P1), while the node-level negative sampling implied by the objective reduces the likelihood of OWA violations (P2). An anchored variant of the cosine distance yields a locality-preserving loss function (P3), and the reconstruction mechanism naturally supports adaptive updates without retraining (P4). Beyond fulfilling these theoretical properties, LORE is computationally efficient and scalable because its single-layer architecture and relational vector representations keep parameter counts stable even in highly multi-relational settings. Despite this lean design, the benchmark experiments show that LORE matches or exceeds the predictive performance of established baselines. Most importantly, it exhibits substantially higher robustness and superior adaptivity. Existing GNN methods, apart from Navi, do not benefit from updated graph topologies without retraining, whereas LORE is able to learn from newly added edges in real time.

These adaptivity results stem from an incompleteness simulation designed to reflect realistic OWA scenarios in which true facts may be missing and added later. While LORE is transductive at the level of base embeddings, it is partially inductive in practice because any new node with known neighbors can obtain an embedding immediately through self-reconstruction, provided those neighbors have base embeddings available. This mixed setting suggests several promising directions for future work. Extending the evaluation to updates involving new nodes without predefined base embeddings, edge deletions, and enlarged metadata such as new relation types could further clarify the limits and strengths of LORE in evolving graphs. Large-scale knowledge graphs such as DBpedia (Auer et al., 2007) or Wikidata (Vrandečić & Krötzsch, 2014), as well as robustness analyses under noisy or adversarial updates, would help assess its applicability at scale. Another open question concerns the extent to which the single-layer reconstruction mechanism captures multi-hop effects and how these can be preserved in dynamic settings, for example through lightweight rule-based inference or occasional retraining of the reconstruction mechanism, without compromising real-time adaptability or property P4. Future work may also explore applications beyond node classification, including link prediction, graph completion, and graph-level reasoning, or integrate LORE into deep architectures by masking base embeddings throughout forward passes.

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

# A APPENDIX

## A.1 EVALUATION DATASETS

The SG benchmarks in Table 4 include citation networks (Cora, CiteSeer, PubMed), where nodes represent papers and edges represent citation links, as well as the Amazon-Computers co-purchase network, where nodes correspond to products and edges indicate frequent co-purchases. These graphs vary in size and label diversity, providing a broad evaluation spectrum for node classification.

| | Nodes | Edges | Classes | Train | Validation | Test |
|---|---|---|---|---|---|---|
| **Cora** | 2,708 | 5,278 | 7 | 140 | 500 | 1,000 |
| **CiteSeer** | 3,327 | 4,552 | 6 | 120 | 500 | 1,000 |
| **PubMed** | 19,717 | 44,324 | 3 | 60 | 500 | 1,000 |
| **Amazon** | 13,752 | 245,861 | 10 | 8,251 | 2,750 | 2,751 |

Table 4: Dataset statistics for undirected graph benchmarks. Nodes and edges refer to the full graphs; Train/Validation/Test indicate the splits used for node classification. Class balances are color-coded.

The KG benchmarks in Table 5 cover diverse domains. AIFB and MUTAG are smaller academic and chemical datasets, while BGS (geological survey data) and AM (museum catalogue) are larger and more complex. The parenthesized values correspond to the original graphs before non-boolean literals were removed, whereas the first values indicate the filtered graphs used in our experiments.

| | Nodes | Relations | Edges | Classes | Train | Test |
|---|---|---|---|---|---|---|
| **AIFB** | 2,835 (8,285) | 22 | 20,264 (29,043) | 4 | 140 | 36 |
| **MUTAG** | 22,506 (23,644) | 18 | 64,494 (74,227) | 2 | 272 | 68 |
| **BGS** | 103,055 (333,845) | 62 | 529,945 (916,199) | 2 | 117 | 29 |
| **AM** | 958,232 (1,666,764) | 45 | 3,299,143 (5,988,321) | 11 | 802 | 198 |

Table 5: Dataset statistics before and after removing non-boolean literals (original values are added in parentheses). Boolean literals are retained as new nodes. Class balances are color-coded.

## A.2 BASELINE CONFIGURATIONS

As baseline models, we consider HOPE, RESCAL, Node2Vec, RDF2Vec, TransE, TransR, TransD, (R)GCN, (R)GAT, and Navi. All embeddings were $\ell^2$-normalized and fixed to a dimensionality of $d = 256$. For GCN- and GAT-based baselines, GEs were first pretrained in a self-supervised manner using standard edge reconstruction with sigmoid cross-entropy loss (Kipf & Welling, 2016). Hyperparameters were selected using the performance of a downstream RF with 100 trees and depth 30. We tuned the number of message-passing layers $1, 2, 3$ and, for GAT/RGAT, the number of attention heads $1, 2$. For KGs, this approach is relation-aware, predicting the validity of directed triples, while for SGs, all edges are treated as undirected and unlabeled. HOPE and RESCAL were tuned over ranks $64, 128, 256$ and regularization strengths $10^{-5}, 10^{-4}, 10^{-3}$ to match embedding dimensionality and prevent overfitting. Training batches for translational models consist of randomly sampled edge subsets of size 256 for SGs, AIFB, and MUTAG, and 512 for BGS and AM, without enforcing connectivity. Hyperparameters for translational models were selected via grid search over MRL margin values $0.5, 0.75, 1.0$. Node2Vec and RDF2Vec were tuned over walk lengths $4, 6, 8$, walks per node $10, 20$, and window sizes $5, 10$. The KG benchmarks do not provide validation splits and thus hyperparameters were selected via nested cross-validation. All baseline models were trained using the Adam optimizer with learning rates selected from $10^{-2}, 5 \cdot 10^{-3}, 10^{-3}, 5 \cdot 10^{-4}, 10^{-4}$, for 100 epochs with early stopping on validation performance. Finally, Navi used TransE embeddings as fixed base GEs for its initial node representations.

### A.3 Additional Evaluation Material

#### A.3.1 Macro-F1 Scores

Tables 6 and 7 report macro-F1 scores for the SG and KG embedding benchmarks, providing a class-balanced view of performance. For all SG datasets, the macro-F1 values mirror the accuracy trends in Table 1: GAT and LoRE achieve the best overall performance, with LoRE showing consistently lower variance. Similarly, the KG results align with the accuracy outcomes in Table 2: RGAT and LoRE lead across most datasets, with LoRE slightly outperforming RGAT on MUTAG, BGS, and matching it on AIFB. These macro-F1 results reinforce that LoRE's performance advantage is consistent across both accuracy and class-balanced evaluation metrics.

| | HOPE | Node2Vec | GCN | GAT | GRACE | LoRE |
|---|---|---|---|---|---|---|
| **Cora** | 80.12 ±1.70 | 80.76 ±1.56 | 81.89 ±1.63 | **83.84** ±2.03 | 83.76 ±1.16 | 83.66 ±**0.99** |
| **CiteSeer** | 68.41 ±1.96 | 69.83 ±1.78 | 69.21 ±1.83 | **71.40** ±2.18 | 70.84 ±1.22 | 71.15 ±**1.11** |
| **PubMed** | 77.85 ±1.79 | 77.52 ±1.63 | 78.60 ±1.70 | 79.62 ±2.10 | **80.70** ±**1.45** | 79.18 ±**1.16** |
| **Amazon** | 84.03 ±1.84 | 84.77 ±1.51 | 85.66 ±1.74 | 86.30 ±2.15 | 86.56 ±1.22 | **86.82** ±**1.08** |

Table 6: Macro-F1 scores including standard deviations for the SG embedding benchmark tasks.

| | RESCAL | TransE | TransR | TransD | RDF2Vec | RGCN | RGAT | LoRE |
|---|---|---|---|---|---|---|---|---|
| **AIFB** | 80.45 ±1.33 | 81.67 ±1.70 | 79.92 ±1.77 | 69.84 ±1.93 | 84.15 ±1.82 | 87.12 ±1.28 | 90.20 ±0.96 | **90.95** ±**0.08** |
| **MUTAG** | 68.72 ±1.61 | 73.49 ±1.86 | 67.82 ±1.79 | 65.93 ±2.02 | 72.04 ±1.89 | 74.11 ±1.54 | 82.10 ±1.14 | **83.45** ±**0.98** |
| **BGS** | 66.05 ±1.74 | 67.63 ±1.95 | 66.54 ±1.97 | 67.95 ±1.89 | 66.80 ±1.99 | 71.05 ±1.44 | 72.18 ±1.17 | **73.01** ±**1.09** |
| **AM** | 70.54 ±1.70 | 68.81 ±1.85 | 70.42 ±1.89 | 76.96 ±1.96 | 75.60 ±1.73 | **82.83** ±1.31 | 78.01 ±**0.92** | 81.53 ±1.01 |

Table 7: Macro-F1 scores including standard deviations for the KG embedding benchmark tasks.

#### A.3.2 Comparison of Forward and Backpropagation Times

To obtain these measurements reported in Figure 2, we simulated forward passes and corresponding backpropagation steps with Adam for $|\mathcal{R}| = 5, 10, \ldots, 50$ relation types. For each configuration, $|\mathcal{R}|$ random node embeddings (embedding dimension 256) were generated, assuming one incoming edge per relation type. Each experiment was repeated ten times for each number of relations and graph embedding model, and averages were recorded. The results demonstrate that the two-layer RGCN and RGAT variants exhibit quasi-linear scaling in computation time, whereas LoRE maintains nearly flat scaling and therefore offers superior efficiency in multi-relational scenarios.

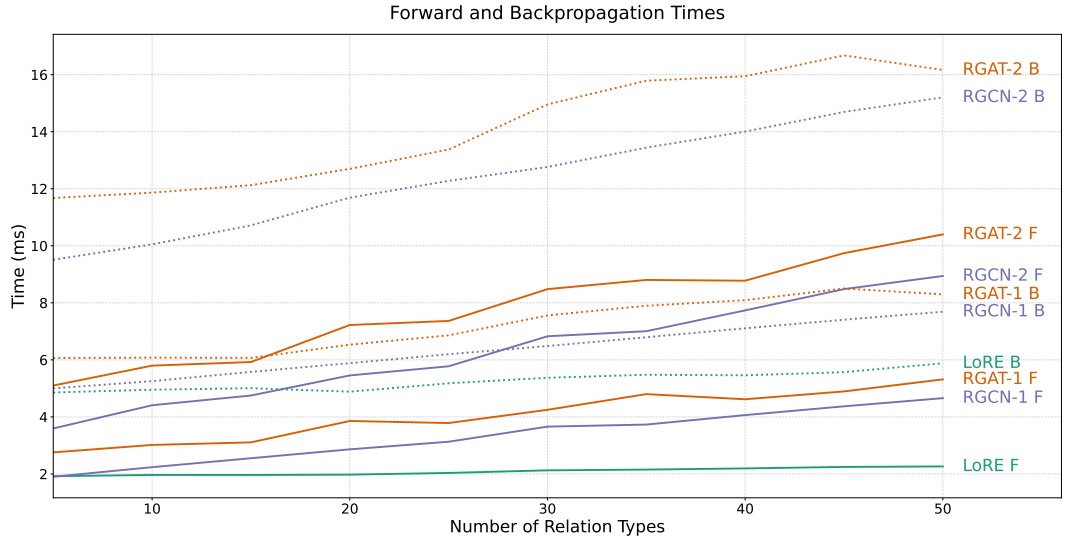

Figure 2: Forward and backpropagation times for LoRE, RGCN, and RGAT models across varying numbers of relation types. Solid lines represent forward passes, while dotted lines represent backpropagation using Adam. For RGCN and RGAT, both one-layer and two-layer variants are shown.

### A.3.3 Comparison of $\lambda_{\text{LoRE}}$ to other Loss Functions

We compare $\lambda_{\text{LoRE}}$ to four baselines obtained by pairing the cosine and Euclidean distances

$$\delta_{\ell_2}(x, x') = \|x - x'\|_2 \quad \text{and} \quad \delta_{\cos}(x, x') = 1 - \frac{x^\top x'}{\|x\|_2 \cdot \|x'\|_2},$$

with either MRL or Bayesian Personalized Ranking (BPR). The MRL objective for a distance $\delta$ is

$$\lambda_{\text{MRL}}^{\delta}(\hat{y}, y, \tilde{y}) = \max\left(\delta(\hat{y}, y) - \delta(\hat{y}, \tilde{y}) + \mu,\ 0\right).$$

For BPR, we operate directly in distance space. To encourage the prediction $\hat{y}$ to be closer to the positive target $y$ than to the negative target $\tilde{y}$, we define

$$\lambda_{\text{BPR}}^{\delta}(\hat{y}, y, \tilde{y}) = -\log \sigma(\delta(\hat{y}, \tilde{y}) - \delta(\hat{y}, y)), \tag{6}$$

where $\sigma(z) = (1 + e^{-z})^{-1}$ is the logistic sigmoid. This yields the baselines $\lambda_{\text{MRL}}^{\ell_2}$, $\lambda_{\text{MRL}}^{\cos}$, $\lambda_{\text{BPR}}^{\ell_2}$, and $\lambda_{\text{BPR}}^{\cos}$. We evaluate all alternatives on the same benchmark tasks and with the same LoRE training configuration as in Section 5.1 and Section 5.2. Since the standard cosine distance is not locality-aware, we tune the MRL margin over $\mu \in \{0.5,\ 1.0,\ 1.5\}$, while for the Euclidean distance we select from $\{0.1,\ 0.25,\ 0.5,\ 1.0\}$. Although the baselines often achieve comparable mean accuracies, $\lambda_{\text{LoRE}}$ typically attains the best results and shows substantially higher stability, making it the preferred loss function for LoRE, even without considering its role in enabling locality awareness (P3). The results are summarized in Table 8 and Table 9.

|  | $\lambda_{\text{MRL}}^{\ell_2}$ | $\lambda_{\text{MRL}}^{\cos}$ | $\lambda_{\text{BPR}}^{\ell_2}$ | $\lambda_{\text{BPR}}^{\cos}$ | $\lambda_{\text{LoRE}}$ |
|---|---|---|---|---|---|
| **Cora** | 84.54 ±1.79 | 82.80 ±1.29 | 84.03 ±1.64 | 83.31 ±1.18 | **84.88 ± 0.95** |
| **CiteSeer** | 71.03 ±1.99 | **73.18**±1.51 | 72.77 ±1.12 | 72.01 ±1.83 | 73.12 ±**1.04** |
| **PubMed** | 80.57 ±1.40 | 79.64 ±1.72 | 81.89 ±1.54 | 81.60 ±2.13 | **82.21 ±1.14** |
| **Amazon** | 88.18 ±1.80 | 87.75 ±1.37 | 87.82 ±1.82 | 88.44 ±1.42 | **88.90 ±0.96** |

Table 8: Mean accuracies including standard deviations of LoRE for the SG embedding benchmark tasks from Section 5.1 depending on the selected loss function.

|  | $\lambda_{\text{MRL}}^{\ell_2}$ | $\lambda_{\text{MRL}}^{\cos}$ | $\lambda_{\text{BPR}}^{\ell_2}$ | $\lambda_{\text{BPR}}^{\cos}$ | $\lambda_{\text{LoRE}}$ |
|---|---|---|---|---|---|
| **AIFB** | 90.31 ±0.45 | **91.67 ±0.00** | 91.11 ±0.36 | 89.94 ±1.02 | **91.67 ±0.00** |
| **MUTAG** | 84.42 ±**0.89** | 84.96 ±1.32 | 85.00 ±0.92 | 85.07 ±1.14 | 85.29 ±0.91 |
| **BGS** | 73.70 ±1.81 | 74.16 ±1.23 | 73.88 ±1.49 | **74.19** ±1.50 | 74.14 ±**1.03** |
| **AM** | 91.89 ±2.27 | 92.45 ±1.85 | 90.83 ±1.39 | 92.71 ±1.14 | **92.93 ±0.96** |

Table 9: Mean accuracies including standard deviations of LoRE for the KG embedding benchmark tasks from Section 5.2 depending on the selected loss function.

## A.4 EMBEDDING CHARACTERISTICS

The visualizations in Figure 3 show that LORE embeddings achieve clearer class separation compared to both TransE and RGAT, with more compact clusters and less overlap. While this effect is most pronounced on the AIFB dataset, similar patterns were observed on other benchmarks, suggesting that LORE consistently yields more structured and discriminative representations. The magnifying glass highlights structurally equivalent nodes from three different classes. Due to LORE's neighborhood-invariance, they are mapped within the majority cluster but positioned near its boundary, which is a desirable outcome for balancing structural consistency with class distinctions.

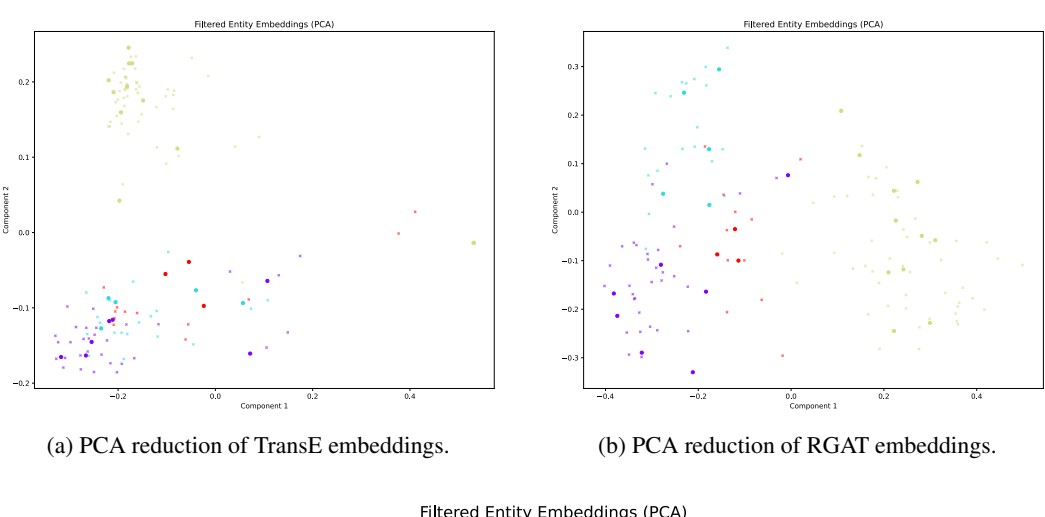

(a) PCA reduction of TransE embeddings.      (b) PCA reduction of RGAT embeddings.

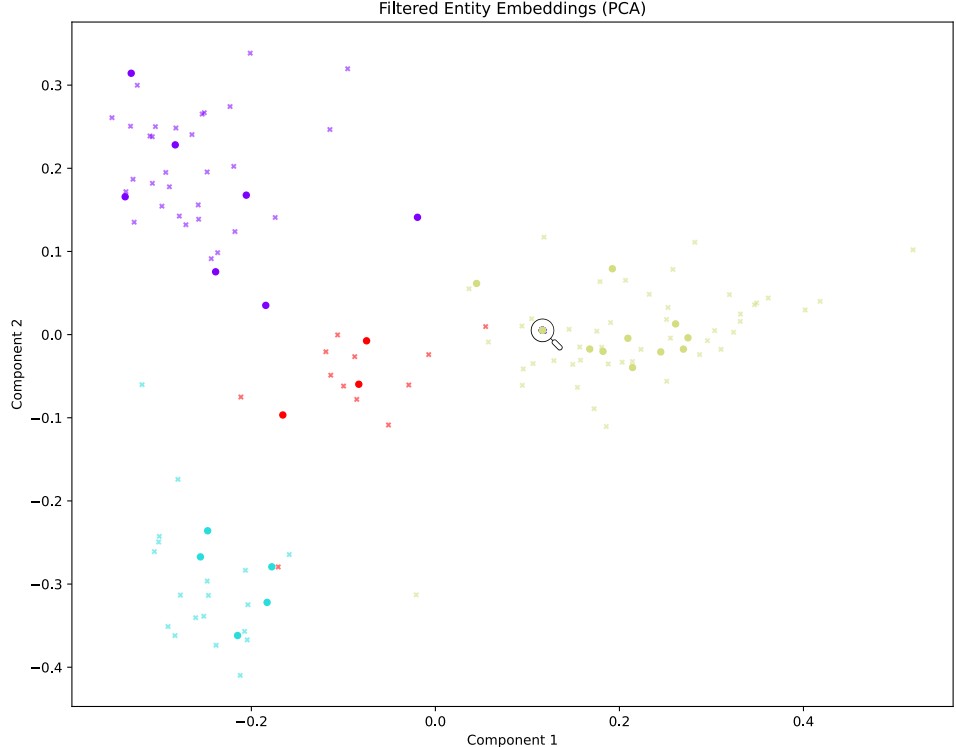

(c) PCA reduction of LORE embeddings.

Figure 3: PCA embedding visualizations of the AIFB task, where four classes are predicted. The PCA plots project the embeddings into 2 dimensions, showing the separability and clustering characteristics for (a) TransE, (b) RGAT, and (c) LORE. Note: Test node embeddings are printed in bold.

