# OpenReview forum: "LoRE: Robust and Adaptive Graph Embeddings via Local Self-Reconstruction Mechanisms"
_ICLR.cc/2026/Conference — Submitted to ICLR 2026_

### Official Review · Reviewer_PW26 · 2025-10-31

**Soundness:** 4
**Presentation:** 4
**Contribution:** 3
**Rating:** 6
**Confidence:** 4

**Summary:**

This paper introduces LORE, a self-supervised graph embedding framework. The authors design it to simultaneously address four key challenges in real-world graph applications: neighborhood-invariance (P1), obedience to the open-world assumption (P2), locality-awareness (P3), and adaptability to graph dynamics (P4). The core mechanism is a model that reconstructs a node's "independent" embedding ($\psi$) using only the base embeddings ($\phi$) of its first-order neighbors. This reconstruction is paired with a node-level contrastive loss, which is designed to respect the OWA under the assumption that it is less likely for two distinct nodes to share an entire neighborhood than a yet unobserved edge. The method's effectiveness is demonstrated on node classification tasks for both simple graphs and knowledge graphs, as well as in a dynamic graph setting.

**Strengths:**

The primary strength is its conceptual design, which directly confronts several practical challenges of graph embedding. More specifically,

1. The reconstruction-based mechanism is inherently inductive, providing an "on-the-fly" method for generating embeddings for new nodes (P4), which is a key requirement for dynamic graphs.
2. By design, the framework enforces neighborhood-invariance (P1), ensuring that structurally identical nodes produce identical embeddings.
3. The shift from traditional edge-level negative sampling to a node-level contrastive loss on full neighborhood reconstructions is a well-reasoned and strong attempt to better respect the OWA (P2).

**Weaknesses:**

W1
The paper would be stronger if it explicitly stated the key assumptions underlying its 1-hop reconstruction mechanism. The method succeeds only if the first-order neighborhood contains sufficient information to define a node's embedding. This is a strong assumption that may not always hold.

W2
I appreciate that the authors attempted to design a loss function that preserves local geometry (P3). However, the justification for the novel and complex 'perspective-preserving' loss ($\delta_{LORE}$) is not fully justified. It is unclear why this anchor-based loss is essential. The paper would benefit significantly from an ablation study that compares $\delta_{LORE}$ directly against more standard contrastive loss functions (e.g., a margin-based cosine similarity or InfoNCE). Without this, it is unclear if the added complexity is necessary or if a simpler, more common objective would achieve similar results.

W3
The empirical results show a very marginal performance gain over the best-performing baselines (e.g., GAT/RGAT). Given the added complexity of the model (e.g., two sets of embeddings per node, a novel loss function), this small improvement in accuracy appears to be insufficient. I am guessing that the small margin is attributed to the easiness of the benchmarks, namely that the baseline models could already achieve near optimal performance and the small improvement by the propose method might be significant but appear to be marginal. I suggest the authors to find challenging benchmarks where baseline models fails but the proposed model works. This reveals more clearly the cases in which the proposed model stands out.

W4
The term "independent node embedding" ($\psi$) is confusing to me. The paper states it is "independent" of the node's own parameters, but it is not independent precisely, i.e., it depends on the base embeddings of its neighbors, and the entire training objective is to make it coherent (i.e., dependent) with the focal node's own base embedding ($\phi(v)$). This choice of terminology makes it unclear, rather than clarifies, the conceptual design of the model.

**Questions:**

Q1: What are the key underlying assumptions about the data for the model to be successful?
Q2: Do we really need every single component to achieve this performance? Is the local geometry preservation really essential to improve the task performances?
Q3: Can you showcase more clearly the cases in which the proposed model stands out?
Q4: Is "independent node embedding" independent? If so, in what sense precisely in math?

---

> ### Author Response · Authors · 2025-11-19
>
> Dear Reviewer PW26,
>
> Thank you very much for the detailed feedback. We carefully revised the paper and
> incorporated your suggestions. Below, we address W1–W4 and Q1–Q4.
>
> W1 (Assumptions of the 1-hop reconstruction mechanism):
>
> We agree that this assumption could have been stated more explicitly. In the revised version,
> we now emphasize that the reconstruction is a 1-hop aggregation. However, at the same
> time, we note that the role of adjacent base embeddings changed. Since they are not used
> inside a node’s own forward pass anymore, they are trained to encode local contextual
> structure and thereby provide the semantic signal needed to reconstruct a neighbor node’s
> embedding. This means that after training, neighbor base embeddings encode contextual
> information from earlier training and can therefore capture aspects of the broader local
> topology. At the same time, we now also explicitly note a limitation: this assumption may
> become brittle under dynamic updates. While LoRE can inductively “reconstruct”
> embeddings for new nodes with known neighbors, large structural modifications that alter
> multi-hop context may require occasional retraining of base embeddings to maintain global
> consistency. We emphasize this caveat in the updated version and discuss it as an important
> direction for future work.
>
> W2 (Justification for the perspective-preserving loss):
>
> In the revised version, we first clarify that $\delta_{LoRE}$ denotes the distance function, which in turn induces the loss $\lambda_{LoRE}$. In response to the request for an ablation, we added a comparison of LoRE’s perspective-preserving loss against well-established alternatives (MRL and BPR combined with standard euclidean and cosine). These results were obtained from the same experimental pipeline as the original submission and are now included in the appendix and referenced in Section 4.3. They show that while $\lambda_{LoRE}$ already provides slightly higher mean accuracy, its main advantage is a substantial reduction in predictive variance, resulting in notably more stable embeddings. This supports its use as the default loss in LoRE.
>
> W3 (Performance interpretation):
>
> We would like to clarify a possible misunderstanding: standard GNNs also maintain two
> embeddings per node (a base embedding and an aggregated embedding, which in LoRE
> corresponds to the reconstruction), so the architectural complexity is identical up to the point
> where message passing begins. Computationally, LoRE remains lean because its layer
> operations use simple vector additions instead of costly matrix multiplications, which
> becomes a significant advantage once multiple GNN layers or many relation types are
> involved, as is typical in real-world graphs. In terms of evaluation, we focus on accuracy,
> stability, and efficiency. While the accuracy gains over strong baselines are indeed modest,
> LoRE consistently matches or exceeds them while being substantially more stable (lower
> variance) and far more efficient. More challenging benchmarks could further highlight these
> advantages, and we now explicitly mention this in the future-work section.
>
> W4 (“Independent node embedding”):
>
> We agree the term could have been misleading. It referred solely to the reconstruction being
> computed without using the node’s own base embedding. The revised paper now clarifies
> that this independence applies only to the reconstruction mechanism and not to the final
> embedding.
>
> Q1:
>
> LoRE relies on the same core assumption shared by virtually all graph embedding models:
> that the graph contains sufficiently meaningful local structure for context-based
> representations to be useful. We do not impose additional assumptions beyond this.
>
> Q2:
>
> The locality-aware loss and the reconstruction mechanism each contribute distinct and
> complementary effects. As discussed in W2, the locality-aware loss does increase mean
> accuracy but, more importantly, substantially improves stability, reducing prediction variance
> compared to standard losses. The reconstructive mechanism is equally necessary, since
> without it LoRE behaves like a standard one-layer encoder and cannot capture richer local
> structure without multi-hop propagation. Together, these components enable LoRE to remain
> competitive in accuracy while being considerably more stable and efficient than the
> baselines.
>
> Q3:
>
> Standard benchmarks make this difficult to showcase because strong baselines already
> perform well. However, LoRE’s advantages became most visible in two situations: (1) cases
> like the AIFB dataset, where mean accuracy is similar but LoRE exhibits much lower
> variance, indicating higher stability and less sensitivity to noise, and (2) dynamic or
> incomplete settings, where LoRE can benefit from newly added edges without retraining,
> unlike conventional GNNs (Section 5.3)
>
> Q4: See W4.

---

> > ### Comment · Reviewer_PW26 · 2025-11-21
> > **Response**
> >
> > Thanks for clarifying. I now understand that LORE's key contribution in terms of benchmark performance is not just mean performance gain, but also (and more crucially) stability and efficiency. The authors also clarify the underlying assumptions about one-hop reconstruction mechanisms, and in hindsight, I see that the LORE encodes the broader context, not just direct neighbors, into the embedding. While very curious to see how actually they really encode the context, I appreciate the authors making this point very clear and discussing it in the conclusion section. I increased my rating.

---

### Official Review · Reviewer_pTb8 · 2025-11-01

**Soundness:** 3
**Presentation:** 3
**Contribution:** 3
**Rating:** 6
**Confidence:** 2

**Summary:**

This paper proposes LoRE, an attention-based “local self-reconstruction” graph embedding framework. Its core idea is to perform self-reconstruction on randomly initialized base embeddings using only the target node's neighborhood and relation types, thereby forming independent embeddings that satisfy neighborhood invariance (P1). Simultaneously, it mitigates violations of the open-world assumption (OWA) (P2) through node-level negative samples and an angle loss preserving perspective, preserves local geometric relationships during training (P3), and enables online adaptation via forward updates without full retraining when graphs are updated (P4). Experiments on simple graphs and knowledge graphs demonstrate that LoRE achieves classification accuracy comparable to or surpassing strong baselines while showcasing its dynamic adaptation capabilities.

**Strengths:**

1. This paper is well-written, with clear motivations and solutions.
2. It achieves accuracy comparable to or better than strong baselines on SG/KG benchmarks, while demonstrating advantages in parameter efficiency and training time across multi-relational scenarios.
3. Dynamic update experiments show that LoRE improves downstream classification with edge updates without requiring full retraining, demonstrating practical applicability.

**Weaknesses:**

1. Dynamic capabilities currently only evaluate “delete-and-reinsert” scenarios; they could be extended to cover additional scenarios such as adding new nodes and removing nodes.
2. Adding references when introducing baselines in Section 5.2 is essential. I am unclear why the authors omitted this, as it prevents readers from determining when these methods were proposed and hinders quick access to their detailed descriptions.
3. The table design lacks clarity, making it difficult to identify the top-ranked and second-ranked methods for each task.

**Questions:**

See Weekness

---

> ### Author Response · Authors · 2025-11-19
>
> Dear Reviewer pTb8,
>
> Thank you very much for your constructive feedback. We incorporated your comments into
> an updated version of the paper. Your remarks have been addressed as follows.
>
> Q1 (Dynamic capabilities):
>
> We now explicitly state in Section 5.3 that the dynamic scenario is implemented
> as simulated graph incompleteness, which is one of the most common causes of graph
> dynamics [1]. Removing edges, training embeddings, reinserting edges, and updating
> embeddings therefore provides a first, controlled evaluation of LoRE’s adaptiveness. We
> fully agree that dynamics extend beyond incompleteness, including cases where nodes are
> added or removed. Since \textsc{LoRE} is transductive at the level of base embeddings but
> inductive in its reconstruction mechanism, fully inductive scenarios such as embedding
> entirely new nodes without any prior context require different evaluation setups. We plan to
> extend LoRE to broader dynamic scenarios in future work, but a full treatment would be out
> of scope for the present paper, which focuses on formally introducing the LoRE framework.
>
> Q2 (Missing references in Section 5):
>
> Most baselines used in our evaluation are already introduced and cited in the Related Work
> section. This was originally done to avoid clusters of citation markers in the evaluation
> section that could harm readability. However, we agree that this may be confusing for
> readers who look at the evaluation first. In the rebuttal version of the paper, we therefore
> added explicit statements in Section 5 clarifying that the baselines are introduced and
> referenced in Related Work. We could also include the references directly in Section 5 in the
> camera-ready version if that presentation is preferred. It is feasible within the page limits.
>
> Q3 (Table clarity):
>
> Thank you for pointing out the difficulty in identifying the strongest methods in the tables.
> Before submission, we discussed various options for emphasizing results, since our
> comparison focuses on both mean accuracy and robustness (such as the lower confidence
> bound). In line with your feedback, we now decided to highlight the highest mean accuracy
> and the lowest standard deviation separately for each task. This makes the ranking clearer.
>
> [1] Schmelzeisen, L., Dima, C., & Staab, S. (2021). Wikidated 1.0: An Evolving Knowledge
> Graph Dataset of Wikidata’s Revision History. In Wikidata@ISWC

---

> > ### Comment · Reviewer_pTb8 · 2025-11-28
> >
> > I can't see your updated version. I manually checked several comparison methods and also reviewed the related work again. Actually, I think these methods may be a bit outdated.

---

> > > ### Author Response · Authors · 2025-11-28
> > >
> > > Thank you for your follow-up. We are equally surprised, as we and at least one other reviewer can access the updated PDF without issues. We checked for possible reasons and, given the timing, this may be related to yesterday's OpenReview data incident. Community discussions indicate that several authors and reviewers reported temporary loss of access to revision data, so it might be possible that this affected your visibility of our rebuttal version uploaded on November 20th.
> > >
> > > A quick way to verify this is to check whether GRACE (Zhu et al., 2020) appears around line 372 in the version you are seeing. We added this baseline in response to another reviewer and it already addresses your concern regarding more current comparison methods. Although GRACE performs strongly, our main findings remain unchanged: LoRE reaches similar or better accuracy while being more efficient and more stable.
> > >
> > > We also clarified more explicitly in the updated version that LoRE is a global, task-agnostic embedding method. Recent work on the benchmark tasks used in our paper has shifted toward task-specific, end-to-end training objectives. These models achieve high performance on a single classification task, but they are not comparable when the goal is to learn general-purpose embeddings. From this perspective, established self-supervised baselines such as (R)GCN or (R)GAT remain relevant and appropriate comparison points. This clarification is included in the revised PDF, but we hope this summary here is helpful in the meantime.
> > >
> > > Regarding your initial comment on referencing baselines in the evaluation section, the updated version now states:
> > >
> > > “Baselines include Node2Vec, GCN, and GAT (see Section 2), as well as HOPE (Ou et al., 2016) [...] we also include GRACE (Zhu et al., 2020) as a strong contrastive benchmark.”
> > >
> > > “Baselines are introduced in Section 2 and include RESCAL, TransE, TransR, TransD, RDF2Vec, RGCN, and RGAT.”
> > >
> > > If the visibility issue persists, we are happy to provide the updated sections directly here. Please let us know if we can supply any part of the revision in the meantime.

---

### Official Review · Reviewer_nWQx · 2025-11-06

**Soundness:** 2
**Presentation:** 2
**Contribution:** 2
**Rating:** 4
**Confidence:** 5

**Summary:**

This paper presents a graph embedding framework called LORE (LOcally Reconstructed Embeddings). The experimental results demonstrate the effectiveness of LORE.

**Strengths:**

S1. This paper claims that LORE enforces neighborhood-invariance, enables learning under incomplete graphs, preserves locality-awareness, and integrates graph updates on the fly.

S2. The experiments show that LORE achieves competitive or superior accuracies compared to some baselines.

**Weaknesses:**

W1. The theoretical part of the paper is not easy to follow. For example, as shown in Definition 5, A= V \times M. Later, A= V \times O. It is better to clearly present the mathematical symbols.

W2. The organization of the paper should be improved. For instance, the section of Related Work in placed on Section 3. Maybe, it will be better to move this section to Section 2.

W3. The experimental results are not convincing. For example, "community detection, node classification, and link prediction" are classical downstream tasks for graph embedding as mentioned in the Abstract. Then, it is better to demonstrate the performance of the proposed methods on these tasks to demonstrate the advantages of the proposed methods. For another example, this paper states that "Incremental updates are simulated by randomly removing roughly half of each test node's edges before training, while ensuring that at least one edge remains per node." in Section 5.3. The authors should clearly prove the correctness of this kind of simulation. Otherwise, it is difficult to understand the results presented in Table 3.

W4. Some statements in this paper are not reasonable. For example, the authors claim that "Simple undirected graphs and KGs are considered as representative graph types." Why these kinds of graph types are considered as representatives?

W5. The baselines should be improved. For example, as shown in Table 1, HOPE (published in 2016), Node2Vec (2016), GCN (2017), and GAT (2018). Why not include more recent work, such as [1]? The authors should give the reasons for the baseline selection rather than just saying that "Baselines include HOPE (Ou et al., 2016), Node2Vec, GCN, and GAT".

[1] Powerful Graph Convolutional Networks with Adaptive Propagation Mechanism for Homophily and Heterophily. AAAI 2022.

**Questions:**

W1-W5

---

> ### Author Response · Authors · 2025-11-19
>
> Dear Reviewer nWQx,
>
> We thank you for the thorough and constructive feedback. Your comments were highly
> helpful in identifying places where the presentation of our paper could be sharpened,
> clarified, reorganized, or improved in general. We are confident that the revised version of
> the paper is significantly improved as a result. Below, we describe how we incorporated your
> feedback for points W1–W5.
>
> W1 & W2 (Notation clarity and organization):
>
> While incorporating your feedback into our rebuttal version, we realized that W1 is largely
> addressed by the structural concern raised in W2. The earlier structure resulted from a
> trade-off we faced before submission: either introduce the preliminaries first so that all
> terminology is formally defined before Related Work, or place Related Work earlier and
> provide only a high-level introduction there. In the end, this decision was essentially a coin
> toss. We opted for preliminaries first, which avoided duplicated definitions but also made it
> easier to overlook early notational inconsistencies.
>
> The inconsistency you pointed out after Definition 5 was indeed an oversight. While
> technically correct, we fully agree that the presentation could be confusing to readers
> encountering our terminology for the first time. Following your suggestion, we reorganized
> the manuscript and moved Related Work in front of the preliminaries. While this
> rearrangement changes the structure of the theoretical sections, the core content remains
> the same. During this restructuring, we carefully reviewed all definitions and notation and
> identified some additional places where clearer terminology and more consistent notation
> improved readability. Obviously, this also affected the remainder of the paper but we believe
> that these changes substantially strengthen both the theoretical foundation and the overall
> flow of our paper.
>
> W3 (Experimental results):
>
> We agree that listing specific downstream tasks such as community detection and link
> prediction in our abstract was misleading, since our evaluation focuses solely on node
> classification. Because LoRE aims to generate task-agnostic embeddings and property P2
> discourages standard link-prediction benchmarks, we rewrote this part of the abstract to
> refer only to downstream applications in general and to explicitly emphasize the
> task-agnostic nature of the embeddings.
>
> Regarding the dynamic simulation in Section 5.3, we now clarified that this setup models
> incompleteness as dynamics in OWA settings, a common real-world scenario [1] where
> edges are not yet present during training and appear later. Removing and reinserting edges
> provides a controlled and reproducible way to test a model's ability to handle such updates.
> While these results are preliminary, we now explicitly outline future work on more complex
> dynamic settings, including inductive node addition, edge deletions, and combinations of
> on-the-fly updates with periodic retraining of the reconstruction mechanism.
>
> W4 (Reasonability of statements):
>
> Regarding the concrete example: As with dynamic updates, the space of possible edge
> metadata is broad. In this work, we restrict ourselves to orientations and edge-type labels,
> which we now state explicitly. Other metadata, such as timestamps or weights, are left for
> future work. Simple graphs and knowledge graphs were chosen as two scenarios that cover
> none (SGs) or both (KGs) of these metadata types and provide established benchmark
> datasets.
>
> In general, we also revised a few statements that may have appeared overly complex or
> insufficiently explained during the restructuring described in W1 & W2.
>
> W5 (Baseline selection):
>
> LoRE is designed to learn global, task-agnostic embeddings that are trained independently
> of any specific prediction task. Many recent approaches, including the one you referenced,
> learn embeddings jointly with a specific downstream objective. As explicitly stated in their
> problem formulations, this approach optimizes embeddings with respect to classification
> tasks and is therefore not directly comparable to LoRE or to classical unsupervised
> baselines such as Node2Vec, HOPE, GCN, or GAT. We now highlight this assumption more
> clearly at the beginning of the paper.
>
> However, to strengthen the evaluation, we additionally included GRACE [2], a strong and
> recent contrastive baseline that produces task-agnostic embeddings. While GRACE
> achieves slightly higher mean accuracy on some datasets, it also shows substantially higher
> variance, which aligns with the previous results.
>
> [1] Schmelzeisen, L., Dima, C., & Staab, S. (2021). Wikidated 1.0: An Evolving Knowledge
> Graph Dataset of Wikidata’s Revision History. In Wikidata@ISWC
>
> [2] Zhu, Y., Xu, Y., Yu, F., Liu, Q., Wu, S., & Wang, L. (2020). Deep Graph Contrastive
> Representation Learning. arXiv:2006.04131.

---

### Author Response · Authors · 2025-12-02
**TLDR for the AC**

LoRE is a framework for learning global and task agnostic graph embeddings through a local reconstruction mechanism. The approach is designed to produce embeddings that are robust and adaptive to incompleteness in graph data.

We summarize our main insights from the rebuttal process up until November 28. Details can be found in the updated PDF and in the discussions with the reviewers.

Reviewer 1 (nWQx):

Reviewer 1 provided detailed comments on structure, notation, and clarity. These remarks led to a reorganization of the paper, adjustments to notation, and a clearer presentation of the theoretical sections. The revised version also states more explicitly that the work focuses on global and task agnostic embeddings. An additional baseline was added to the benchmark section as suggested by this reviewer. No further comments were received after the updated version was uploaded.

Reviewer 2 (pTb8):

Reviewer 2 reported not being able to view the updated PDF. All remarks from the initial review were incorporated into the revision, including clearer referencing of baselines and additional clarifications in the evaluation section. The follow up comment was posted on November 28, shortly before the initial ICLR notification. This may explain the visibility issue. The follow up question aligns with a point raised by Reviewer 1 that was addressed in the updated version.

Reviewer 3 (PW26):

Reviewer 3 requested a clearer explanation of how LoRE reconstructions capture contextual information. The revised version provides additional clarification of this mechanism and its assumptions. The remaining comments, including those related to terminology and ablation studies, were also addressed. In the final discussion, the reviewer noted that the revisions resolved the concerns and the score was increased accordingly.

---

### Meta-Review · Area_Chair_VjPP · 2025-12-29

**Summary:**

This presents a self-supervised graph embedding framework called LORE (LOcally Reconstructed Embeddings). The methods is based on a local reconstruction mechanism that performs self-reconstruction on randomly initialized embedding using target node's neighbohood, leading to independent embeddings that satisfy neighborhood invariance, mitigates open world assumption with a contrastive approach with local geometric relationships geometries during training and allows online adaptation when the graphs are updated. Experiments show the effectiveness of the methods on node classification tasks for simple graphs and knowledge graphs in static and dynamic settings.

-Reviewer nWQx identifies as strengths the neighborhood invariance enabling learning under incomplete graphs, preserving locality-awareness, and integrating graph updates on the fly. The experiments show that the method is competitive or superior to the compared baselines.
On the other hand, theoretical part of the paper is not easy to follow, organization of the paper could be improved, experimental results are not convincing : misses community detection, link prediction tasks and the dynamic setup is not justified. Lack of justification of the types of graphs used. The baselines considered can be improved,  the ones used are mainly before 2019.

-pTb8 mentions as strengths that the paper is well-written, with clear motivations and solutions, it achieve comparable or better accuracy with strong baselines with more efficiency in parameter and training time, improvements in dynamic settings.
On the other hand, Dynamic capabilities are only evaluated on "delete-and-reinsert" scenarios, Table design likes clarity, missing references in Section 5.2. The reviewer mentions that he finds the baselines a bit outdated.

-PW26 mentions as strengths the conceptual design allowing on-the-fly method for generating embeddings for new nodes, allowing also neighborhood-invariance and capacity to deal with open-world assumption.
Among the weaknesses, the reviewer indicated the limitation due to the need of having all the needed information in the first-order neighborhood, lack of ablation study for the "perspective-preserving" loss, benchmark choice and discussion on limited improvement.


The paper is borderline. On the one hand, authors provide clear clarifications on the method leading to a better understanding of the approach and its capabilities. On the other hand I retain the limitations raised regarding the experiments with rather old baselines and the need of a strong revision which would need another round of revision.
Based on these last two elements, I recommend rejection.

**Reviewer Concerns:**

-Reviewer nWQx received answers for each of these remarks,  regarding clarification and presentation issues authors proposed a reorganization of the paper, adjustments of notations and review of the presentation of thereotical part , this requires in my opinion another round of reviews since authors acknowledged some misleading or imprecise claims. An additional baseline was added, but answers on the limitations of the experiments remain limited.

-For pTb8, authors provided an answer on the dynamic settings and proposed an update to improve the clarity of the Table. They added the missing references and completed their evaluation with an additional baseline. I am more skeptical on the experimental part since authors added only another one baseline of 2020 (similar answer as for nWQx) while the reviewer mentioned that he found the proposed beelines were before 2019. The reviewer mentioned that he was not able to see the revision, but I do not think that this changes significantly the evaluation.

-For PW26, the answers provided by the authors helped to better understand the method and to clarify the weaknesses raised (ablation study, terminology, contextual information), the reviewer acknowledged the relevance of the answers.

**Reviewer Scores:**

-nWQx gave a 4, given the limited answers on the experimental part and the rather large revision, I am not convinced that the reviewer would increase his score.

-pTb8 gave a 6, issues related to clarity and the dynamic settings were I guess addressed. This is not completely the case for the experimental part, si I do not think that the reviewer would have increased his score.

-PW26 gave a 6 and indicated that he was willing to increase his score.

---

### Decision · Program_Chairs · 2026-01-26

Reject